# Diving into Class-Incremental Learning from Better Balancing Old and New knowledge

## Abstract

Class-Incremental Learning (Class-IL) aims to continuously learn new knowledge without forgetting old knowledge from a given data stream using deep neural networks. Recent Class-IL methods strive to balance old and new knowledge and have achieved excellent results in mitigating the forgetting by mainly employing the rehearsal-based strategy. However, the representation learning on new tasks is often impaired since the trade-off is hard to taken between old and knowledge. To overcome this challenge, based on the Complementary Learning System (CLS) theory, we propose a novel CLS-based method by focusing on the representation of old and new knowledge in Class-IL, which can acquire more new knowledge from new tasks while consolidating the old knowledge so as to make a better balance between them. Specifically, our proposed method has two novel components: (1) To effectively mitigate the forgetting, we first propose a bidirectional transport (BDT) strategy between old and new models, which can better integrate the old knowledge into the new knowledge and meanwhile enforce the old knowledge to be better consolidated by bidirectionally transferring parameters across old and new models. (2) To ensure that the representation of new knowledge is not impaired by the old knowledge, we further devise a selective momentum (SMT) mechanism to give parameters greater flexibility to learn new knowledge while transferring important old knowledge, which is achieved by selectively (momentum) updating network parameters through parameter importance evaluation. Extensive experiments on four benchmarks show that our proposed method significantly outperforms the state-of-the-arts under the Class-IL setting.

## 1 Introduction

Catastrophic forgetting (McCloskey & Cohen, 1989), i.e., learning new knowledge while forgetting previously learned old knowledge, is a long-standing problem for continual learning. The Class-Incremental Learning (Class-IL) setting, as the setting closest to real-world application scenarios in continual learning, has been widely studied through various strategies, e.g., regularization-based strategy (Kirkpatrick et al., 2017; Zenke et al., 2017; Li & Hoiem, 2017; Aljundi et al., 2018), architecture-based strategy (Rusu et al., 2016; Mallya & Lazebnik, 2018; Serra et al., 2018) and rehearsal-based strategy (Lopez-Paz & Ranzato, 2017; Chaudhry et al., 2018; Belouadah & Popescu, 2019; Zhou et al., 2021; Caccia et al., 2022; Kang et al., 2022; Wang et al., 2022; Zhou et al., 2022).

Recent studies strive to balance old and new knowledge while mitigating the catastrophic forgetting mainly with the rehearsal-based strategy under the Class-IL setting. A line of works (Rebuffi et al., 2017; Buzzega et al., 2020; Boschini et al., 2022) resort to simple knowledge distillation (Hinton et al., 2015; Sarfraz et al., 2021), i.e., they preserve old knowledge by aligning the output logits of current new model (student model) with the output logits of previous models (teacher model) and adjust loss weights to balance old and new knowledge. Inspired by the Complementary Learning System (CLS) theory (Kumaran et al., 2016; Singh et al., 2022), another line of works (Pham et al., 2021; 2022; Arani et al., 2022; Sarfraz et al., 2023b;a) combine the distillation with CLS theory: a short-term model is built for fast learning the episodic knowledge and a long-term model is built for slow learning the general structured knowledge by simulating the two learning systems 'hippocampus' and 'neocortex' in the brain, so that the old and new knowledge can be better obtained. These methods have made great progress in balancing old and new knowledge by adjusting the loss weights or proposing effective strategies. However, the trade-off is hard to taken between old and

new knowledge, resulting in that these methods are not stable enough to represent both the old and new knowledge well. Particularly, the representation of new knowledge is often impaired due to the excessive integration of the retained old knowledge under the Class-IL setting.

To address this problem, we propose a novel CLS-based method by focusing on the representation of old and new knowledge in Class-IL, termed BDT-SMT, which can acquire more new knowledge from new tasks while consolidating the old knowledge so as to make a better balance between them. Specifically, similar to (Arani et al., 2022), our proposed method (see Fig. 1) has three main modules: a working model and two semantic memory models (i.e., the long-term and short-term models that simulate the two learning systems of the brain). Built on this, we first devise a bidirectional transport (BDT) strategy to transfer parameters directly and bidirectionally between the working model and the two semantic memory models. We denote the transport process (working model → semantic memory models) as backward transport and that (semantic memory models → working model) as forward transport, respectively. This is quite different from (Arani et al., 2022; Sarfraz et al., 2023b;a) with only one unidirectional process (i.e., backward transport). With the BDT strategy, our proposed method forms a circular transport channel among these three models to transfer information to each other more smoothly, thus effectively mitigate the forgetting. Note that the extension to bidirectional transport is *not that easy*: (1) Only one unidirectional process is concerned even in the latest works (Sarfraz et al., 2023b;a). (2) Within the CLS framework, the bidirectional process becomes challenging across three models. (3) The forward transport of bidirectional process may impair the representation of new knowledge, which is also the reason why the selective momentum (SMT) mechanism is carefully designed along with BDT in this paper.

Furthermore, to ensure that the representation of new knowledge is not impaired by the (integrated) old knowledge, we devise a selective momentum (SMT) mechanism to selectively (momentum) update parameters with the evaluation of parameter importance during the forward transport. Concretely, a parameter importance evaluation algorithm like SI (Zenke et al., 2017) is introduced into SMT, and an importance threshold is set to control the momentum updates of the parameters, so as to receive more important knowledge from old tasks for important parameters while giving greater flexibility to other unimportant parameters for better learning new tasks. Thus, the designed SMT mechanism in our method is able to enforce the model to continuously transfer important old knowledge as well as learn new knowledge significantly better through selective momentum updating of network parameters. Note that the biggest parameter-updating difference between SI (or EWC (Kirkpatrick et al., 2017)) and our BDT-SMT lies in that the parameters of our method are *selectively momentum* updated by the parameters of old model according to parameters importance, while all parameters of SI/EWC are gradient updated by backpropagation according to parameters importance. Additionally, we choose to devise the importance evaluation algorithm according to SI, instead of other strategies (Kirkpatrick et al., 2017; Aljundi et al., 2018), because it is more complementary to our proposed method, which has been shown in Appendix A.3.

Our main contributions are four-fold: **(1)** We propose a novel CLS-based method termed BDT-SMT to acquire more new knowledge from new tasks while consolidating the old knowledge so as to make a better balance between them under the Class-IL setting. **(2)** To effectively mitigate the forgetting, we devise a bidirectional transport (BDT) strategy between old and new models, which is quite different from the latest works (Arani et al., 2022; Sarfraz et al., 2023b;a) with only one unidirectional process (i.e., backward transport). Moreover, to ensure that the representation of new knowledge is not impaired by the old knowledge during forward transport, we design a selective momentum (SMT) mechanism to selectively (momentum) update network parameters through parameter importance evaluation. **(3)** Extensive experiments on four benchmarks show that our proposed method significantly outperforms the state-of-the-art methods under the Class-IL setting. **(4)** The proposed BDT and SMT have a high flexibility/generalizability, which can be widely applied to not only continual learning but also (momentum-based) contrastive learning.

## 2 METHODOLOGY

### 2.1 PROBLEM DEFINITION

The Class-IL setting is concerned for continual learning, where the model is trained for image classification on a sequential tasks $T$, i.e., $T = \{T_1, T_2, ..., T_H\}$, where $H$ denotes the number of tasks. For each task $T_t$ ($1 \leq t \leq H$) from $T$, it owns a task-specific training set $D_t = \{(x_i^t, y_i^t)\}_{i=1}^{N_t}$ with $N_t$ sample pairs, where $x_i^t \in R^D$ is a sample image from the class $y_i^t \in Y^t$. $Y^t$ is the label space

of the task $T_t$. For label spaces of different tasks $T_t$ and $T_{t'}$, there is non-overlapping classes, i.e., $Y^t \cap Y^{t'} = \emptyset$ for $t \neq t'$. Further, for each task $T_t$, its validation and test sets can be defined similarly. The goal of Class-IL is to enforce the trained model to accurately classify all previously seen classes with less forgetting after learning all tasks without providing the task identifiers.

## 2.2 BASE FRAMEWORK

Similar to the recent work (Arani et al., 2022), the base framework of our BDT-SMT has three main modules: a working model and two semantic memory models (i.e., plastic model and stable model, see Fig. 1). These three models are deployed with the same network structure (e.g., ResNet18 (He et al., 2016) as the backbone) and initialized with the same parameters. The main functions of these modules are described separately below.

**Working Model**  The function of working model is two-fold: on one hand, it is trained to continuously learn the incoming tasks on a given data stream ($D$) to acquire new knowledge; on the other hand, it needs to continuously transfer the learned knowledge to the semantic memory models to accumulate the old knowledge. Concretely, given the current batch data $(X_D, Y_D)$ from the dataset $D_t$ and the replayed batch data $(X_M, Y_M)$ randomly sampled from the memory buffer $M_t$ for the task $T_t$ as the inputs, the working model $F(; \theta_W)$ is trained to accurately classify the current task data and replayed old data by backpropagation (see Eq. (6) for the total loss). Meanwhile, the working model continuously transfers the learned knowledge (information) stored in $\theta_W$ to the two semantic memory models, i.e., it uses the parameters $\theta_W$ to update the parameters of semantic memory models, so that the semantic memory models can obtain the long-term and short-term memory retention of knowledge (see below for the detailed transfer process). Thus, the working model plays an important role (updating & transferring parameters) in the framework.

**Semantic Memory Models**  The semantic memory models, i.e., stable model (S) and plastic model (P), are maintained to retain the knowledge previously learned on the working model. The stable model $F(; \theta_S)$ continuously accumulates the knowledge as long-term memory to acquire the slow adaptation of information, while the plastic model $F(; \theta_P)$ continuously accumulates the knowledge as short-term memory to acquire the fast adaptation of information. The realization of long-term period and short-term period mainly depends on the update frequency of parameters. Thus, instead of updating parameters at each training step, the frequency parameters are employed on semantic memory models to stochastically control their updates. Concretely, the long-term stable model is used to retain more information of earlier tasks with a small frequency parameter $f_S$, while the short-term plastic model is used to retain more information of recent tasks with a big frequency parameter $f_P$ ($f_P \geq f_S$). Compared with the short-term memory model, the long-term memory model can accumulate more general structured knowledge over time, leading to better generalization across tasks. Therefore, the stable model is used for inference to achieve optimal performance.

Given the parameters of working model $\theta_W$, the parameters of plastic model $\theta_P$ and the parameters of stable model $\theta_S$, the parameters of two semantic memory models $\theta_P, \theta_S$ are (momentum) updated by the parameters of working model $\theta_W$ through the Exponential Moving Average (EMA) strategy (Tarvainen & Valpola, 2017; Grill et al., 2020), which are formulated as:

$$
\begin{aligned}
\theta_P &= m_P \cdot \theta_P + (1 - m_P) \cdot \theta_W, \quad \text{if rand(1)} < f_P \\
\theta_S &= m_S \cdot \theta_S + (1 - m_S) \cdot \theta_W, \quad \text{if rand(1)} < f_S
\end{aligned}
\tag{1}
$$

where $m_P$ and $m_S \in [0, 1)$ denote the backward transport momentum parameters, and rand(1) denotes a random value sampled from a standard Gaussian distribution. By setting $m_P \leq m_S$, the plastic model can adjust new information faster (more), while the stable model can adjust new information slower (less) so as to construct the general structured knowledge over time. Note that the parameters of working model $\theta_W$ are updated by backpropagation, and the parameters of two semantic memory models are only updated by $\theta_W$ since they have no gradients.

## 2.3 OUR BDT-SMT METHOD

In this paper, built on the above base framework, our BDT-SMT devises two novel components, i.e., bidirectional transport (BDT) strategy and selective momentum (SMT) mechanism. The combination of BDT and SMT facilitates the model to acquire more new knowledge from new tasks while consolidating the old knowledge, thereby achieving a better balance between old and new knowledge and significantly improving model performance.

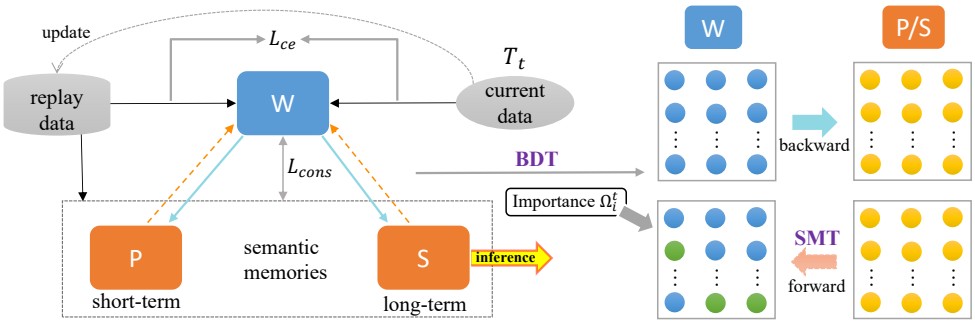

Figure 1: The overview of our proposed method built over the working model (W) and two semantic memory models (plastic model (P) and stable model (S)).

**Bidirectional Transport (BDT) Strategy** To effectively mitigate the forgetting, we propose a BDT strategy (including forward & backward transport) to transfer information more smoothly between the three models in the form of a circular transport channel ($P \Leftrightarrow W \Leftrightarrow S$), as shown in Fig. 1. Compared to the unidirectional transport (only backward transport) used in (Arani et al., 2022), the BDT strategy indeed strengthens the information communication among the three models: it enables the knowledge of earlier tasks (from stable model), the knowledge of recent tasks (from plastic model) and the knowledge of new tasks (from working model) to be continuously transferred and flowed between models, thereby better consolidating the old knowledge and mitigating the forgetting. Concretely, at each training step, the parameters of working model ($\theta_W$) are first updated by backpropagation. Then, with the updated working model, it is required to determine whether to momentum update the plastic model and stable model according to Eq. (1) (i.e., backward transport). Let $U_{PS}$ denote the set of semantic memory models that are updated by the working model ($U_{PS}$ may be $\emptyset$, $\{P\}$, $\{S\}$, and $\{P,S\}$). In turn, the updated semantic memory models are used to momentum update the working model at the beginning of the next training step (i.e., forward transport). Formally, given the parameters of the three models $\theta_W, \theta_P$ and $\theta_S$, the momentum update of the parameters of the working model in forward transport adopts the same EMA as in Eq. (1) with the forward transport momentum parameter $\hat{m} \in [0, 1)$:

$$\theta_W = \hat{m} \cdot \theta_W + (1 - \hat{m}) \cdot \theta_j, \ \ j \in U_{PS}. \tag{2}$$

In our experiments, we find that the plastic model ($P$) must be used for momentum updating before the stable model ($S$) when $U_{PS} = \{P, S\}$. And note that the forward transport begins with the second task to ensure that semantic memory models have accumulated knowledge. The deployment of BDT enables the model to effectively mitigate the forgetting, which can be observed from Table 3: the average forgetting is reduced by 8.54% on S-CIFAR-10 and 4.74% on S-CIFAR-100, respectively (see Base+BDT vs. Base), where (Arani et al., 2022) is denoted as Base. Furthermore, the combination of BDT and SMT yields significant improvements over (Arani et al., 2022), which can also be observed from Table 3: the average accuracy increases by 4.0% and 2.66%, and the average forgetting drops by 9.47% and 2.5% on S-CIFAR-10 and S-CIFAR-100, respectively (see Base+BDT+SMT vs. Base). The proposed SMT mechanism will be detailed below.

**Selective Momentum (SMT) Mechanism** Although good retention of old knowledge and promising improvement of model performance are obtained with the BDT strategy, there still exists a clear limitation in the representation of new knowledge. Specifically, in the forward transport of BDT, all parameters of the working model are momentum updated by the parameters of the semantic memory models. Among these parameters of semantic memory models, some parameters are helpful (positive) to the representation of new knowledge, such as parameters representing general structured knowledge; while some parameters are useless or even harmful (negative) to the representation of new knowledge, resulting in that the new knowledge is not well represented. In this paper, we thus design a SMT mechanism to ensure that the representation of new knowledge is not impaired by the integrated old knowledge, which is achieved by selectively updating parameters with the evaluation of parameters importance during forward transport, as shown in Fig. 1.

The SMT mechanism is designed to give greater flexibility to unimportant parameters of previous tasks for better learning new tasks. Concretely, we first need to make an evaluation to the importance of parameters of working model by the importance evaluation algorithm, then rank the parameter importance and set an importance threshold $k$ ($k$ is defined as a fraction here) to indicate the parameter range that needs to be updated. As a result, the Top-$k$ part of parameters in the working

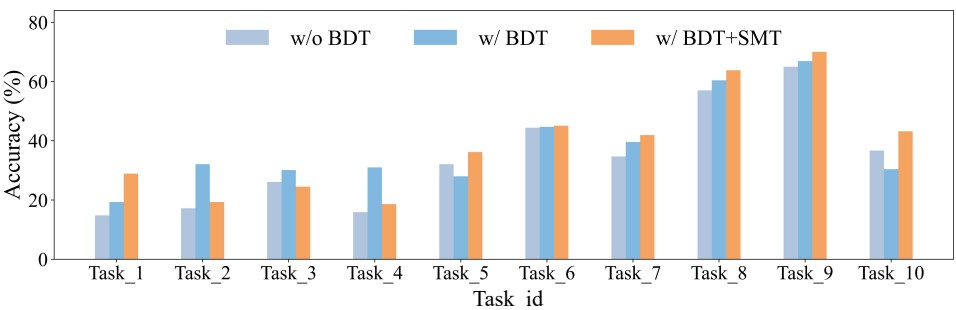

Figure 2: Comparative results of the final model on ten learned tasks of S-CIFAR-100 among the base framework without BDT ("w/o BDT"), the base framework with BDT ("w/ BDT"), and the base framework with BDT+SMT ("w/ BDT+SMT").

model that are important to previous tasks are momentum updated by the corresponding Top-$k$ part of parameters of semantic memory models, and other unimportant parameters of working model are no longer updated by the semantic memory models (see Fig. 4 for the selection of $k$). Let $V_{top}$ denote the set of important Top-$k$ part of parameters. The forward transport is reformulated as:

$$\theta_W[i] = \hat{m} \cdot \theta_W[i] + (1 - \hat{m}) \cdot \theta_j[i], \;\; j \in U_{PS}, \;\; i \in V_{top}, \tag{3}$$

where $\theta_W[i]$ (or $\theta_j[i]$) is the $i$-th element of $\theta_W$ (or $\theta_j$). With this mechanism, our BDT-SMT can receive more important knowledge from old tasks for important parameters while giving greater flexibility to unimportant parameters for fitting new tasks.

For the importance evaluation algorithm, it is devised according to SI (Zenke et al., 2017). Differently, in SI, the parameters importance is used as a penalty strength in the loss function to enforce all parameters to be gradient updated by backpropagation; in our BDT-SMT, the parameters importance is used as a criterion to determine whether the parameter needs to be updated, so that the parameters can be *selectively momentum* updated according to the threshold $k$. Formally, when the importance of parameter $\theta_W[i]$ for task $t$ ($T_t$) is denoted as $\omega_i^t$, we can approximately define it as the contribution of $\theta_W[i]$ to the change of the total loss throughout the training phase of task $t$:

$$\omega_i^t \equiv \sum_s \eta \cdot g_{s,t}^2(\theta_W[i]), \tag{4}$$

where $\eta$ is the learning rate, and $g_{s,t}(\theta_W[i])$ is the gradient of $\theta_W[i]$ at the training step $s$ for task $t$. Notably $\omega_i^t$ only measures the absolute contribution of the parameter, ignoring its own update range. Thus, a normalized importance (regularization strength) $\Omega_i^t$ for task $t$ is given by:

$$\Omega_i^t = \sum_{\tau \leq t} \frac{\omega_i^\tau}{(\triangle_i^\tau)^2 + \varepsilon}, \;\; \triangle_i^\tau \equiv \theta_W^\tau[i] - \theta_W^{\tau-1}[i], \tag{5}$$

where task $\tau$ denotes a task before or current task $t$. $\triangle_i^\tau$ denotes the update amount of $\theta_W[i]$ in task $\tau$ ($\theta_W^\tau$ denotes $\theta_W$ at the end of task $\tau$). $\varepsilon$ is a small constant used to prevent calculation instability. Note that $\omega_i^t$ is initialized to zero at the beginning of each task, while $\Omega_i^t$ is only initialized to zero at the beginning of the first task and updated by the accumulated $\omega_i^t$ at the end of each task. The pseudocode of importance evaluation algorithm is shown in Appendix A.2.

By deploying SMT to selectively update the parameters of working model, the transfer of harmful old parameters is effectively avoided, enforcing the working model to learn richer representation of new knowledge. As a result, under the joint action of our BDT and SMT, the stable model is able to obtain more and better general structured knowledge over time, which facilitates better generalization across tasks (i.e., the old and new tasks), resulting in superior performance. To make this clearer, we compare the accuracy results of the final model (obtained from one independent run) on ten learned tasks of S-CIFAR-100 among the base framework without BDT, with BDT, and with BDT+SMT in Fig. 2. We can see that: (1) Compared with "w/o BDT", the utilization of BDT enforces the model to better consolidate old knowledge on earlier tasks and meanwhile achieve better results on recent tasks. This finding highlights that the knowledge transfer among three models plays a crucial role in facilitating the smoothing of the decision boundaries of new and old tasks (i.e., greatly reducing the distribution overlap between old and new classes), thereby effectively mitigating the forgetting. Meanwhile, the knowledge transfer also brings benefits to new tasks, further demonstrating its importance in continual learning. (2) The utilization of SMT

empowers the model to achieve higher accuracies on recent and new tasks. This directly shows that the model's representation ability in new tasks has been substantially improved by applying SMT, thereby facilitating the acquisition of better general structured knowledge over time.

**Total Loss** The total loss ($L$) is composed of a cross entropy loss ($L_{ce}$) and a consistency loss ($L_{cons}$). Among them, the cross entropy loss is computed on the current batch data ($X_D, Y_D$) and replayed batch data ($X_M, Y_M$), due to its simpleness, we mainly introduce the consistency loss here. Specifically, the consistency loss is computed on the replayed batch data ($X_M, Y_M$) by aligning the output logits of working model ($Z'_W = F(X_M; \theta_W)$) with the optimal output logits of semantic memory models ($Z_P = F(X_M; \theta_P)$ for plastic model or $Z_S = F(X_M; \theta_S)$ for stable model). The optimal output logits are determined by an optimal strategy, which is expressed as a better representation of semantic information for the replay data between plastic model and stable model, i.e., the final selected semantic logits own higher softmax scores for the ground-truth labels of the inputs. Therefore, the total loss function $L$ is formulated as:

$$L = L_{ce}(sf(Z_W, Y)) + \gamma L_{cons}(Z'_W, Z), \tag{6}$$

$$Z_W = F((X_D \cup X_M); \theta_W), \tag{7}$$

where $Z_W$ is the output logits of two batch data $X_D$ and $X_M$; $Z'_W$ is the output logits of the replayed batch data $X_M$; $Z$ denotes the optimal output logits of $X_M$, i.e., $Z_S$ or $Z_P$. $Y$ refers to the ground-truth labels of $X_D$ and $X_M$, i.e., $Y = Y_D \cup Y_M$. $sf(\cdot)$ represents the softmax function, $\gamma$ represents the balancing hyperparameter. Note that $L_{cons}$ is defined as a Mean Squared Error (MSE) loss.

## 3 EXPERIMENTS

### 3.1 EXPERIMENTAL SETUP

**Datasets** Four standard benchmark datasets are used to evaluate the model performance under the Class-IL setting. (1) **S-MNIST** is obtained by splitting digit-base dataset MNIST (LeCun et al., 2010) into 5 consecutive tasks with two classes per task. For each class in the task, there are 6,000 images for training and 1,000 images for testing. These images are gray-scale images with the resolution $28 * 28$. (2) **S-CIFAR-10** is obtained by splitting the dataset CIFAR-10 (Krizhevsky et al., 2009) into 5 consecutive tasks with two classes per task. For each class in the task, there are 5,000 images for training and 1,000 images for testing. The resolution of these color images is $32*32*3$. (3) **S-CIFAR-100** is obtained by splitting the dataset CIFAR-100 (Krizhevsky et al., 2009) into 10 consecutive tasks with 10 classes per task. For each class in the task, there are 500 images for training and 100 images for testing. These images have the same resolution as S-CIFAR-10. (4) **S-Tiny-ImageNet** is obtained by splitting the dataset Tiny-ImageNet (Banerjee & Iyer, 2015) into 10 consecutive tasks with 20 classes per task. For each class in the task, there are 500 images for training and 50 images for testing. The resolution of these color images is $64 * 64 * 3$. Note that a fixed order for all classes is kept in each dataset for sequential training across ten independent runs. The implementation details are provided in Appendix A.2. The source code will be released soon.

**Evaluation Metrics** To evaluate the model performance under the Class-IL setting, average accuracy (Acc) and average forgetting (Fg) (Buzzega et al., 2020; Fini et al., 2022; Pham et al., 2022) are reported after learning all tasks across ten independent runs. For the accuracy, it refers to the accuracy of the last task on the previous tasks, where a larger value indicates a better model performance (main metric for continual learning). For the forgetting, it refers to the difference between the accuracy of the last task on previous task and the acquired maximum accuracy on this task, where a smaller value indicates a better model performance. Formally, given the number of tasks $H$ and the accuracy $a_{i,j}(j \leq i)$ of the task $i$ on the previous task $j$, as in previous works, average accuracy Acc ($\uparrow$) and average forgetting Fg ($\downarrow$) can be formulated as follows:

$$Acc(\uparrow) = \frac{1}{H} \sum_{j=1}^{H} a_{H,j}, \quad Fg(\downarrow) = \frac{1}{H-1} \sum_{j=1}^{H-1} \max_{\tau \in 1, \dots H-1} a_{\tau,j} - a_{H,j}. \tag{8}$$

### 3.2 MAIN RESULTS

Table 1 shows the comparative results w.r.t. the state-of-the-arts in terms of average accuracy on the four benchmark datasets. Following (Arani et al., 2022), five baselines are used as competitors: ER (Riemer et al., 2018), GEM (Lopez-Paz & Ranzato, 2017), iCaRL (Rebuffi et al., 2017), DER++ (Buzzega et al., 2020) and CLS-ER (Arani et al., 2022). Furthermore, four latest methods

Table 1: Comparison to the state-of-the-arts under the Class-IL setting in terms of average accuracy over ten independent runs. The standard deviation is given in brackets. All methods (with the same backbone) are trained from scratch.

| Method | Buffer Size | S-MNIST | S-CIFAR-10 | S-CIFAR-100 | S-Tiny-ImageNet |
|---|---|---|---|---|---|
| JOINT (upper bound) | – | 95.57 (±0.24) | 92.20 (±0.15) | 70.55 (±0.91) | 59.99 (±0.19) |
| SGD (lower bound) | – | 19.60 (±0.04) | 19.62 (±0.05) | 9.32 (±0.06) | 7.92 (±0.26) |
| ER (Riemer et al., 2018) | 200 | 80.43 (±1.89) | 44.79 (±1.86) | 14.78 (±0.67) | 8.49 (±0.16) |
| GEM (Lopez-Paz & Ranzato, 2017) | 200 | 80.11 (±1.54) | 25.54 (±0.76) | 13.34 (±0.43) | – |
| iCaRL (Rebuffi et al., 2017) | 200 | 70.51 (±0.53) | 49.02 (±3.20) | 35.99 (±0.49) | 7.53 (±0.79) |
| ER-ACE (Caccia et al., 2022) | 200 | 85.24 (±0.65) | 64.08 (±1.68) | 27.85 (±0.61) | 12.73 (±0.66) |
| DER++ (Buzzega et al., 2020) | 200 | 85.61 (±1.40) | 64.88 (±1.17) | 26.40 (±1.17) | 10.96 (±1.17) |
| X-DER (Boschini et al., 2022) | 200 | – | 65.51 (±1.82) | 35.83 (±0.53) | 19.98 (±0.76) |
| CLS-ER (Arani et al., 2022) | 200 | 89.54 (±0.21) | 66.19 (±0.75) | 35.39 (±1.15) | 23.47 (±0.80) |
| SCoMMER (Sarfraz et al., 2023b) | 200 | – | 67.87 (±0.47) | 31.75 (±1.39) | 16.61 (±0.46) |
| ESMER (Sarfraz et al., 2023a) | 200 | 89.21 (±0.26) | 68.51 (±0.33) | 35.72 (±0.25) | 23.37 (±0.11) |
| BDT-SMT (ours) | 200 | **89.99 (±0.27)** | **70.19 (±1.13)** | **38.05 (±0.25)** | **25.31 (±0.29)** |

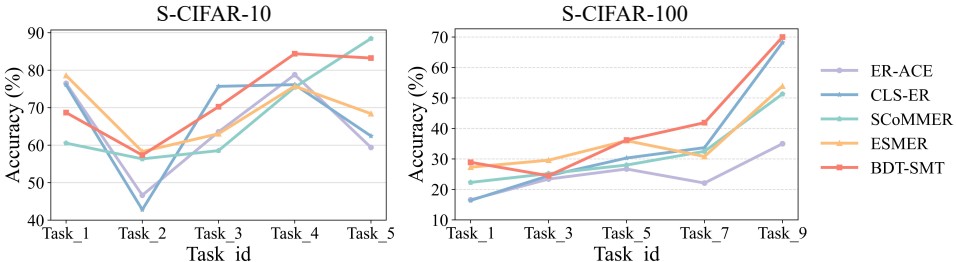

Figure 3: Detailed accuracy comparative results obtained by the final model on each task over the two datasets S-CIFAR-10 and S-CIFAR-100.

are included as additional competitors: ER-ACE Caccia et al. (2022), X-DER Boschini et al. (2022), SCoMMER (Sarfraz et al., 2023b) and ESMER (Sarfraz et al., 2023a). Regarding the five baselines, except S-CIFAR-100 on which we re-implement all of them, the results on other datasets are directly reported from (Arani et al., 2022). As for the four latest methods, we re-implement them on all datasets with their released code. Note that JOINT is the upper bound which indicates that the data from all tasks are used for training together instead of sequential training, while SGD is the lower bound which indicates that all tasks are sequentially trained with fine-tuning.

From Table 1, we can observe that: **(1)** Compared with the state-of-the-arts, our BDT-SMT shows superior performance by achieving the highest average accuracy across all datasets, indicating the effectiveness of our BDT-SMT for Class-IL. **(2)** Our BDT-SMT outperforms the second best ES-MER by an average value 1.98% on three out of four datasets (except S-MNIST), and outperforms the third best CLS-ER by an average value 2.24% on all four datasets. Particularly, our BDT-SMT surpasses ESMER by 2.33% on S-CIFAR-100 and 1.94% on S-Tiny-ImageNet. Additionally, BDT-SMT surpasses CLS-ER by 4.0% on S-CIFAR-10 and 2.61% on S-CIFAR-100. The obtained results provide a direct evidence that our BDT-SMT effectively acquires more new knowledge while consolidating the old knowledge, which yields significant benefits in enhancing model performance. Moreover, Fig. 3 shows detailed comparative results obtained by the final model on each task (take S-CIFAR-10 and S-CIFAR-100 as examples). This clearly demonstrates the effectiveness of our BDT-SMT, i.e., it can make a better balance between old and new knowledge.

To further show the outstanding ability of our BDT-SMT, we conduct experiments under more strict and challenging scenarios: smaller memory buffer sizes ($|M|$, $|M| = 200 \rightarrow 100, 50$) and longer task sequences ($H$, $H = 10 \rightarrow 20$). The comparative results on S-CIFAR-100 and S-Tiny-ImageNet are shown in Table 2. The five comparative methods include ER (Riemer et al., 2018), DER++ (Buzzega et al., 2020), CLS-ER (Arani et al., 2022), SCoMMER (Sarfraz et al., 2023b) and ESMER (Sarfraz et al., 2023a). We implement these methods and ours by adopting the same experimental hyperparameters as in Table 1. It can be observed that: our BDT-SMT consistently achieves optimal results on both datasets under the two scenarios, indicating the superior generalization ability of our BDT-SMT under the Class-IL setting. This ability allows the model to maintain excellent performance even with smaller buffer sizes and longer task sequences.

Table 2: Comparative results obtained by changing the memory buffer size ($|M| = 200 \rightarrow 100, 50$) or the length of task sequences (Tasks Num) ($H = 10 \rightarrow 20$) on S-CIFAR-100 and S-Tiny-ImageNet respectively. The average accuracy is reported over ten independent runs.

| Method | S-CIFAR-100 | | S-Tiny-ImageNet | |
|---|---|---|---|---|
| Buffer Size (with $H$=10 fixed) | $|M|$=50 | $|M|$=100 | $|M|$=50 | $|M|$=100 |
| ER (Riemer et al., 2018) | 10.23 (±0.30) | 11.80 (±0.18) | 8.11 (±0.08) | 8.18 (±0.08) |
| DER++ (Buzzega et al., 2020) | 13.16 (±0.32) | 14.80 (±1.71) | 8.75 (±1.09) | 10.42 (±0.44) |
| CLS-ER (Arani et al., 2022) | 22.80 (±0.48) | 27.91 (±0.65) | 14.34 (±0.58) | 17.53 (±0.88) |
| SCoMMER (Sarfraz et al., 2023b) | 15.48 (±0.40) | 23.61 (±1.20) | 5.28 (±0.71) | 10.33 (±0.55) |
| ESMER (Sarfraz et al., 2023a) | 21.70 (±1.00) | 28.22 (±0.89) | 13.74 (±0.90) | 18.12 (±0.23) |
| BDT-SMT (ours) | **24.45 (±0.63)** | **31.08 (±0.63)** | **15.63 (±0.51)** | **19.33 (±0.70)** |
| Tasks Num (with $|M|$=200 fixed) | $H$=10 | $H$=20 | $H$=10 | $H$=20 |
| ER (Ratcliff, 1990) | 14.78 (±0.67) | 14.61 (±0.49) | 8.49 (±0.16) | 4.82 (±0.19) |
| DER++ (Buzzega et al., 2020) | 26.40 (±1.17) | 19.30 (±1.08) | 10.96 (±1.17) | 8.75 (±0.77) |
| CLS-ER (Arani et al., 2022) | 35.39 (±1.15) | 22.19 (±1.90) | 23.47 (±0.80) | 15.99 (±0.88) |
| SCoMMER (Sarfraz et al., 2023b) | 31.75 (±1.39) | 23.52 (±0.48) | 16.61 (±0.46) | 11.21 (±0.05) |
| ESMER (Sarfraz et al., 2023a) | 35.72 (±0.25) | 27.25 (±0.52) | 23.37 (±0.11) | 10.86 (±0.69) |
| BDT-SMT (ours) | **38.05 (±0.25)** | **28.11 (±0.38)** | **25.31 (±0.29)** | **18.00 (±0.52)** |

Table 3: Ablative results for our BDT-SMT on S-CIFAR-10 and S-CIFAR-100.

| Method | S-CIFAR-10 | | S-CIFAR-100 | |
|---|---|---|---|---|
| | Acc ($\uparrow$) | Fg ($\downarrow$) | Acc ($\uparrow$) | Fg ($\downarrow$) |
| Base (CLS-ER) (Arani et al., 2022) | 66.19 (±0.75) | 29.01 (±3.25) | 35.39 (±1.15) | 35.58 (±1.35) |
| Base+BDT | 67.82 (±1.78) | 20.47 (±5.49) | 36.89 (±0.67) | **30.84 (±1.52)** |
| Base+BDT+SMT (ours) | **70.19 (±1.13)** | **19.54 (±3.60)** | **38.05 (±0.25)** | 33.08 (±0.52) |

## 3.3 ABLATION STUDY

To demonstrate the impact of proposed novel components on the performance of our BDT-SMT, we conduct ablative experiments on S-CIFAR-10 and S-CIFAR-100. The proposed novel components are the BDT strategy and SMT mechanism applied in our BDT-SMT. We thus take the originally followed method CLS-ER (Arani et al., 2022) as the baseline, which is denoted as Base. On the basis of Base (CLS-ER), we first add the BDT strategy, which is denoted as Base+BDT. Then, we add the SMT mechanism, which is denoted as Base+BDT+SMT (i.e., our full BDT-SMT).

The ablative results are shown in Table 3. It can be clearly seen that: **(1)** When the BDT strategy is applied, the average accuracy is improved over Base, and especially the forgetting is greatly reduced (8.54% on S-CIFAR-10 and 4.74% on S-CIFAR-100). These gains strongly prove the effectiveness of the BDT strategy in retaining old knowledge and mitigating the forgetting. **(2)** When the SMT mechanism is also applied, a further improvement in average accuracy is observed (2.37% on S-CIFAR-10 and 1.16% on S-CIFAR-100) in comparison to Base+BDT. The improvement shows the effectiveness of SMT in better acquiring new knowledge, which contributes to the overall enhancement of model performance. Meanwhile, we can see that the average forgetting on S-CIFAR-100 increases, which may be due to lower forgetting caused by the lower maximum accuracy obtained on the previous tasks when only BDT is applied (see Eq. (8)). It is important to notice that the average accuracy is the primary metric to measure the continual learning performance. Furthermore, compared with Base, the average accuracy and average forgetting are significantly improved and decreased respectively (improved by 4.0% for Acc and decreased by 9.47% for Fg on S-CIFAR-10, improved by 2.66% for Acc and decreased by 2.5% for Fg on S-CIFAR-100), demonstrating that the proposed BDT and SMT have significant contributions to the improvement of model performance. Moreover, these results show the complementary of the two proposed components, which is highly instructive for developing more advancing continual learning methods.

Considering the core role of both BDT and SMT, we conduct experiments on S-CIFAR-10 to thoroughly analyze the influence of two crucial hyperparameters, namely $\hat{m}$ and $k$, on the performance of our BDT-SMT. We keep other hyperparameters fixed, to explore the forward transport momentum parameter $\hat{m} \in \{0.9, 0.99, 0.999, 0.9999\}$ and the threshold parameter $k \in \{1, \frac{1}{2}, \frac{1}{3}, \frac{1}{4}\}$. Fig. 4 shows the comparative results for average accuracy (Acc) and average forgetting (Fg) across ten independent runs with different hyperparameters $\hat{m}$ and $k$, respectively. It can be clearly seen that: **(1)** When $\hat{m}$ is 0.999, our BDT-SMT obtains the highest average accuracy and the lowest average forgetting. When $\hat{m}$ is too small or too large, average accuracy and average forgetting tend to be

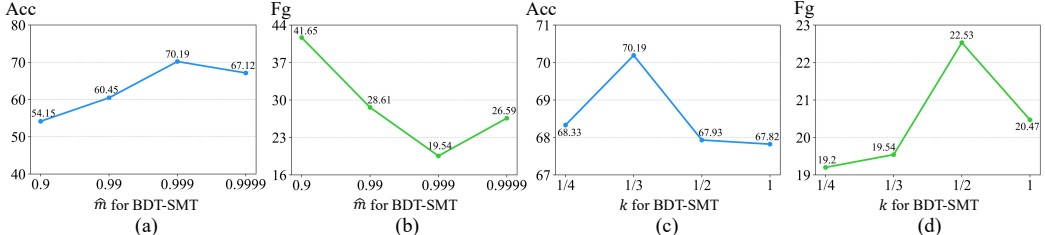

Figure 4: Comparative results of our BDT-SMT with different hyperparameters $\hat{m}$ (a-b, with $k = \frac{1}{3}$ fixed) and $k$ (c-d, with $\hat{m} = 0.999$ fixed) on S-CIFAR-10.

Table 4: Comparative results of SMT-Mean and Mean-ER using a single semantic memory model under the Class-IL setting. The average accuracy is reported over ten independent runs.

| Method | Buffer Size | S-MNIST | S-CIFAR-10 | S-CIFAR-100 | S-Tiny-ImageNet |
|---|---|---|---|---|---|
| JOINT (upper bound) | – | 95.57 (±0.24) | 92.20 (±0.15) | 70.55 (±0.91) | 59.99 (±0.19) |
| SGD (lower bound) | – | 19.60 (±0.04) | 19.62 (±0.05) | 9.32 (±0.06) | 7.92 (±0.26) |
| Mean-ER | 200 | 88.32 (±0.65) | 61.88 (±2.43) | 29.45 (±1.17) | 17.68 (±1.65) |
| SMT-Mean | 200 | **89.33** (±0.36) | **64.06** (±1.32) | **31.22** (±0.88) | **23.68** (±0.39) |
| BDT-SMT | 200 | **89.99** (±0.27) | **70.19** (±1.13) | **38.05** (±0.25) | **25.31** (±0.29) |

compromised since transferring too much information of old parameters would affect the learning of new knowledge, and transferring too little would affect the consolidation of old knowledge. **(2)** When $k$ is $\frac{1}{3}$, our BDT-SMT achieves the best overall performance even if average forgetting is slightly higher compared with $k = \frac{1}{4}$. Thus, we set the forward transport momentum parameter $\hat{m} = 0.999$ on almost all benchmarks, and set the threshold parameter $k = \frac{1}{3}$ on all benchmarks.

### 3.4    FURTHER EVALUATION

To further validate the effectiveness of our proposed components (i.e., the BDT strategy as well as the SMT mechanism) for improving model performance under the Class-IL setting, we conduct extra experiments by employing a single semantic memory model (denoted as SMT-Mean), and compare its performance against the baseline Mean-ER described in (Arani et al., 2022), which also utilizes a single semantic memory model. Table 4 shows the comparative results between SMT-Mean and Mean-ER on all four benchmarks. It can be observed that: our SMT-Mean significantly outperforms Mean-ER in all cases. More importantly, SMT-Mean can match or even exceed the performance of some classic methods in Table 1 (e.g., iCaRL (Rebuffi et al., 2017), and DER++ (Buzzega et al., 2020)). Particularly, the accuracy of our SMT-Mean slightly exceeds that of CLS-ER/ESMER (Sarfraz et al., 2023a) on S-Tiny-ImageNet. These results fully demonstrate the effectiveness of proposed two components, which are extremely beneficial/instructive for continual learning. Furthermore, our BDT-SMT exhibits superior performance over SMT-Mean on all benchmarks, indicating that the complementary application of two semantic memory models is more beneficial to enhancing the model performance compared to using only a single one.

## 4    CONCLUSION

In this paper, we have proposed a novel CLS-based method termed BDT-SMT to acquire more new knowledge from new tasks while consolidating old knowledge so as to make a better balance between them under the Class-IL setting. To effectively mitigate the forgetting, we first devise a bidirectional transport strategy between old and new models, which is quite different from the latest works (Arani et al., 2022; Sarfraz et al., 2023b;a) with only one unidirectional process (i.e., backward transport). Moreover, to ensure that the representation of new knowledge is not impaired by the old knowledge during forward transport, we design a selective momentum mechanism to selectively update parameters with the evaluation of parameters importance. Extensive experiments on four benchmark datasets demonstrate that our BDT-SMT significantly outperforms the state-of-the-arts under the Class-IL setting. Since the proposed BDT and SMT have a high flexibility/generalizability, we will explore how to apply them to other continual learning settings and even (momentum-based) contrastive learning settings in our ongoing research.

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

# A  APPENDIX

## A.1  RELATED WORK

**Knowledge Distillation**   Knowledge distillation (Hinton et al., 2015) is essential in the rehearsal-based methods (Ratcliff, 1990; Riemer et al., 2018; Hou et al., 2019; Buzzega et al., 2020; Zhou et al., 2021; Caccia et al., 2022), which mainly consolidates the old knowledge and mitigates forgetting by distilling knowledge on the replayed data. For example, earlier work iCaRL Rebuffi et al. (2017) mitigates forgetting by replaying the exemplars for nearest-mean-of-exemplars classification and aligning the output scores of the previous step with the current step for distillation. Recent work Dark Experience Replay (DER) Buzzega et al. (2020) mitigates forgetting by matching the network's logits of samples at the current step with the logits of previous steps, where samples are replayed from the memory buffer. DER++ Buzzega et al. (2020) differs from DER in that the replayed samples are also trained with the current new data for classification except for distillation. Most recently, as a modified version of DER, X-DER Boschini et al. (2022) is proposed, the difference is that X-DER includes a regularly update memory buffer by inserting secondary information ('future past') and a future preparation to incoming tasks. Although X-DER improves the model performance compared to DER, it suffers from a very long training time, hindering its practical application for other datasets. The above methods are all simple applications of knowledge distillation.

**CLS Theory**   Recently, inspired by the CLS theory, a line of methods (Pham et al., 2021; 2022; Arani et al., 2022; Sarfraz et al., 2023b;a) are proposed to mitigate forgetting with a combination of distillation and CLS theory. The CLS Kumaran et al. (2016); Singh et al. (2022) theory proposes that there are two interacting systems in the brain: a fast learning system 'hippocampus' encodes new information rapidly as short-term memory, then transfers the acquired information (experience) to the 'neocortex'. The slow learning system 'neocortex' slowly acquires structural information as long-term memory, leading to the construction of semantic knowledge over time. Based on the CLS theory, DualNet Pham et al. (2021) is proposed to mitigate forgetting by using a supervised network as fast net for supervised learning and an unsupervised network as slow net for self-supervised learning with the samples replayed from memory buffer. As an improved version, DualNet++ Pham et al. (2022) introduces a regularization strategy between fast net and slow net to prevent the co-adaptation between them. The recent work CLS-ER Arani et al. (2022) builds long-term and short-term semantic memory models inspired by the CLS theory, and employs the experience replay strategy to align the output logits between the old and new models. Similar methods also include the latest works SCoMMER (Sarfraz et al., 2023b) and ESMER (Sarfraz et al., 2023a), which are both proposed based on the CLS theory. Among them, SCoMMER (Sarfraz et al., 2023b) proposes a semantic dropout mechanism that simulates the sparse coding idea of the brain, enforcing the model to have similar activation units for semantically similar inputs while reducing overlap for semantically dissimilar inputs. ESMER (Sarfraz et al., 2023a) simulates the brain's idea of learning more information from small errors, and proposes a modulation mechanism based on error sensitivity to help the model learn more information. These methods have achieved promising results in balancing old and new knowledge, but they are not stable enough to represent both the old and new knowledge well such as the representation of new knowledge is often impaired (see Fig. 3).

## A.2  IMPLEMENTATION DETAILS

For fair comparisons, our BDT-SMT builds on the framework of CLS-ER (Arani et al., 2022) without modification to the backbone (i.e., a fully-connected network with two hidden layers for S-MNIST, and ResNet18 (He et al., 2016) without pretraining for other three datasets), and adopts its reported experimental settings including batch size, minibatch size, training epochs, backward transport parameter for all benchmarks. The memory buffer size is set to 200, and the reservoir sampling Vitter (1985) is adopted as the updating strategy for memory buffer to ensure sampling with the same probability. To select the optimal hyperparameters for our BDT-SMT, we perform the grid-search strategy on a small validation set of each dataset. During training, Stochastic Gradient Descent (SGD) optimizer is adopted with the learning rate $\eta = 0.1$ for S-MNIST/S-CIFAR-10 and $\eta = 0.05$ for S-CIFAR-100/S-Tiny-ImageNet. For the forward transport momentum parameter $\hat{m}$, we set $\hat{m} = 0.99$ for S-MNIST and $\hat{m} = 0.999$ for S-CIFAR-10/S-CIFAR-100/S-Tiny-ImageNet. For the threshold $k$ used in SMT, we set it to $\frac{1}{3}$ for all benchmarks. Detailed hyperparameter settings are provided in Appendix A.5. Additionally, the pseudocode of the algorithm that computes the normalized importance of parameters at the end of each task is given in Alg. 1.

---

**Algorithm 1** Normalized Importance Evaluation

---

**Input:** current model $F^t(\cdot)$, current data $D_t$, learning rate $\eta$
current replay data for previous tasks $M_t$
the normalized importance $\Omega_i^{t-1}$ of $\theta_W^{t-1}[i]$ at the end of task $t-1$
**Output:** the learned normalized importance $\Omega_i^t$
**Initialization:** $\omega_i^t \leftarrow 0$, $D \leftarrow D_t \cup M_t$, $\theta_W^t[i] \leftarrow \theta_W^{t-1}[i]$
  **for** $1..N$ epochs **do**
    **for** $(x, y)$ in $D$ **do**
      Compute the loss $L$ by Eq. (6)
      Compute the gradient $g(\theta_W^t[i]) = \nabla_{\theta_W^t[i]} L$ by backpropagation
      Compute the updated parameters $\theta_W^t[i] \leftarrow \theta_W^t[i] - \eta \cdot g(\theta_W^t[i])$
      Compute the updated importance $\omega_i^t$ by Eq. (4)
    **end for**
  **end for**
  Update $\triangle_i^t \leftarrow \theta_W^t[i] - \theta_W^{t-1}[i]$, and then update $\Omega_i^t \leftarrow \Omega_i^{t-1} + \frac{\omega_i^t}{(\triangle_i^t)^2 + \varepsilon}$
  **return** the updated normalized importance $\Omega_i^t$

---

Table 5: Average accuracy comparison results over three independent runs on random classes order.

| Method | Buffer Size | S-CIFAR-10 | S-CIFAR-100 |
|---|---|---|---|
| CLS-ER (Arani et al., 2022) | 200 | 64.04 (±1.27) | 35.34 (±0.41) |
| SCoMMER (Sarfraz et al., 2023b) | 200 | 66.38 (±0.65) | 29.31 (±0.41) |
| ESMER (Sarfraz et al., 2023a) | 200 | 66.45 (±1.01) | 36.27 (±0.42) |
| BDT-SMT (ours) | 200 | **66.94** (±0.65) | **37.69** (±0.49) |

Table 6: Comparative results under the Domain-IL setting.

| Method | Buffer Size | R-MNIST | P-MNIST |
|---|---|---|---|
| CLS-ER (Arani et al., 2022) | 200 | 91.99 (±0.54) | 83.65 (±0.28) |
| ESMER (Sarfraz et al., 2023a) | 200 | 90.80 (±0.49) | 81.74 (±0.30) |
| BDT-SMT (ours) | 200 | **92.83** (±0.27) | **87.13** (±0.24) |

### A.3 MORE EXPERIMENTAL RESULTS

**Random Classes Order** Table 5 shows the comparative results when using a random classes order for sequential training (all experiments in the main paper are conducted with a fixed classes order). Specifically, we first obtain a random classes order by creating a pseudo-random number generator using a random seed, and then use the randperm function to ensure that the random classes order is the same for each independent run. We conduct the experiments (buffer size is 200) on the two datasets including S-CIFAR-10 ($H = 5$) and S-CIFAR-100 ($H = 10$), and make comparisons with the-state-of-the-arts including CLS-ER (Arani et al., 2022), SCoMMER (Sarfraz et al., 2023b) and ESMER (Sarfraz et al., 2023a). The experimental parameter settings are adopted as the same with the experiments in the main paper. Here, we set the random seed to 0 for two datasets. From Table 5, it can be seen that our BDT-SMT achieves the highest average accuracy on both datasets. Especially for the dataset S-CIFAR-100, our BDT-SMT significantly outperforms the second-best ESMER. These results provide additional evidence of the effectiveness of our method, highlighting the beneficial impact of the two novel components we have devised: the BDT strategy and the SMT mechanism for continual learning. These two components enable our BDT-SMT to consistently maintain a stable and superior performance even in scenarios with random classes order.

**Domain Incremental Learning Setting** We conduct experiments to investigate the performance of our proposed BDT-SMT under the domain incremental learning (Domain-IL) setting. The Domain-IL setting is usually used to deal with problems with the same label space ($Y$) but different input distributions ($X$), e.g., cats in cartoons and cats in reality. Table 6 shows the comparative results on the two datasets Rotated MNIST (R-MNIST) (Lopez-Paz & Ranzato, 2017) and Permuted MNIST (P-MNIST) (Kirkpatrick et al., 2017), following CLS-ER (Arani et al., 2022). The avarage accuracy is reported over three independent runs. The comparative methods include the state-of-the-art methods CLS-ER and ESMER (Sarfraz et al., 2023a). We implement CLS-ER using the published hyperparameters, while for ESMER, we fine-tune the hyperparameters as they are not provided. From Table 6, we can observe that our BDT-SMT achieves the best performance on both datasets.

Table 7: Comparative results with different PIE algorithms.

| Method | Buffer Size | S-CIFAR-10 | S-CIFAR-100 |
|---|---|---|---|
| PIE-MAS | 200 | 66.11 (±0.25) | 36.34 (±0.62) |
| PIE-EWC | 200 | 68.01 (±0.68) | 36.77 (±0.10) |
| PIE-SI (ours) | 200 | **70.19** (±1.13) | **38.05** (±0.25) |

Table 8: Computational cost comparison results between our BDT-SMT and CLS-ER.

| Method | Compute Time (s) | | Storage (MiB) | |
|---|---|---|---|---|
| | S-CIFAR-10 | S-CIFAR-100 | S-CIFAR-10 | S-CIFAR-100 |
| CLS-ER (Arani et al., 2022) | 8216 | 14626 | 2667 | 1167 |
| BDT-SMT | 8680 | 14874 | 2745 | 1211 |

Table 9: Performance (average accuracy) of three models in BDT-SMT for the main experiments.

| Dataset | Buffer Size | Stable Model | Working Model | Plastic Model |
|---|---|---|---|---|
| S-MNIST | 200 | 89.99 (±0.27) | 89.88 (±0.35) | 89.96 (±0.20) |
| S-CIFAR-10 | 200 | 70.19 (±1.13) | 53.43 (±1.55) | 65.52 (±1.84) |
| S-CIFAR-100 | 200 | 38.05 (±0.25) | 19.09 (±0.37) | 22.83 (±0.48) |
| S-Tiny-ImageNet | 200 | 25.31 (±0.29) | 9.87 (±0.29) | 17.54 (±0.49) |

Specifically, the BDT-SMT outperforms ESMER by an average margin of 3.71% and surpasses CLS-ER by an average margin of 2.16% on the two datasets. These results fully demonstrate the outstanding performance of our BDT-SMT under the Domain-IL setting.

**Comparison with Other Parameter Importance Evaluation Algorithms** In the main paper, we devise the Parameter Importance Evaluation (PIE) algorithm according to SI (Zenke et al., 2017) for experiments. Furthermore, other works such as Elastic Weight Consolidation (EWC) (Kirkpatrick et al., 2017) and Memory Aware Synapses (MAS) (Aljundi et al., 2018) have also introduced their respective PIE algorithms. Among them, EWC calculates the parameter importance by estimating the diagonal values of the Fisher Information Matrix. MAS uses the sensitivity of the output function to estimate the parameter importance. To verify that the PIE algorithm designed according to SI is more applicable to our BDT-SMT, we introduce the PIE algorithms designed according to EWC/MAS into our method for comparative analysis. For better distinction, we denote these methods as PIE-EWC, PIE-MAS and PIE-SI (ours) respectively. For fair comparisons, these comparative methods adopt the same experimental parameter settings as ours. Table 7 shows the average accuracy comparison results over ten independent runs. It can be observed that the PIE-SI achieves the best average accuracy on both datasets, exhibiting a significant performance advantage over the other methods. This outcome provides strong evidence for the superior applicability of the PIE algorithm designed according to SI in our work.

**Computational Cost** We mainly conduct a computational cost comparison between our BDT-SMT and the followed method CLS-ER (Arani et al., 2022), focusing on two key aspects: the average computation time per task (Compute Time) and the total memory storage requirements (Storage). Table 8 shows the comparative results on the S-CIFAR-10 and S-CIFAR-100. It is evident that the computational cost of our BDT-SMT is comparable to that of CLS-ER on each dataset, which further shows the effectiveness of our method at a similar computational cost.

## A.4 PERFORMANCE ANALYSIS

In the BDT-SMT, the stable model is used for inference since it obtains more and better representation of general structured knowledge over time through the transferring and flowing of knowledge between models ($P \Leftrightarrow W \Leftrightarrow S$). Here, we list the average performance (i.e., average accuracy over ten independent runs) of three models in BDT-SMT for the main experiments (i.e., Table 1), as shown in Table 9. It can be observed that the stable model has stronger generalization ability across tasks and thus achieves the best average performance on the sequential tasks of all benchmarks.

Moreover, we show the task-wise performance of our BDT-SMT on the S-CIFAR-10 in Fig. 5, where the results are obtained from one run randomly selected from ten independent runs. After learning one or more tasks on the dataset, the resulting model is evaluated on the test set of the previous tasks. From Fig. 5, it can be seen that: (1) The stable model accumulates and consolidates more knowledge of earlier tasks, and also learns more knowledge of recent and new tasks. (2) The plastic

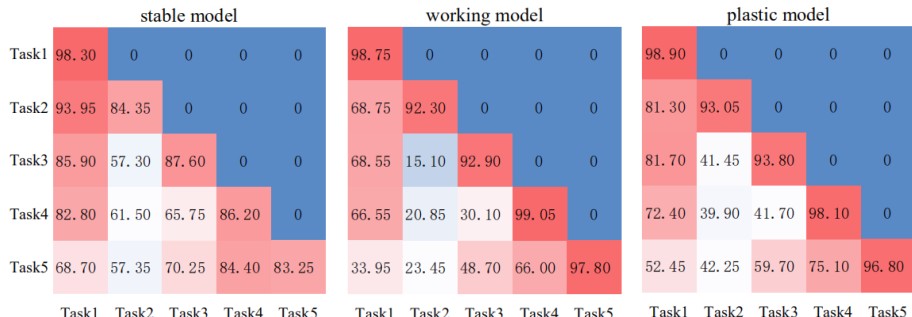

Figure 5: The task-wise performance of our BDT-SMT on S-CIFAR-10 test set.

Table 10: Hyperparameter settings of BDT-SMT and SMT-Mean. 'U'/'L' refers to uniform/longtail.

| Dataset | Buffer Size | BDT-SMT | | | | | | SMT-Mean | | | | |
|---|---|---|---|---|---|---|---|---|---|---|---|---|
| | | $bs$ | $\gamma$ | $\eta$ | $f_S$ | $f_P$ | $\hat{m}$ | $bs$ | $\gamma$ | $\eta$ | $f$ | $\hat{m}$ |
| S-MNIST | 200 | 10 | 2.0 | 0.1 | 0.9 | 1.0 | 0.99 | 10 | 2.0 | 0.1 | 1.0 | 0.99 |
| S-CIFAR-10 | 200 | 32 | 0.15 | 0.1 | 0.1 | 0.6 | 0.999 | 32 | 0.15 | 0.1 | 0.3 | 0.999 |
| S-CIFAR-100 | 200 | 32 | 0.15 | 0.05 | 0.1 | 0.3 | 0.999 | 32 | 0.15 | 0.05 | 0.2 | 0.999 |
| S-Tiny-ImageNet | 200 | 32 | 0.1 | 0.05 | 0.04 | 0.08 | 0.999 | 32 | 0.1 | 0.05 | 0.06 | 0.999 |
| P-MNIST | 200 | 128 | 1.0 | 0.2 | 0.8 | 1.0 | 0.99 | – | – | – | – | – |
| R-MNIST | 200 | 128 | 0.75 | 0.2 | 1.0 | 1.0 | 0.999 | – | – | – | – | – |

Table 11: Hyperparameter settings of SCoMMER (Sarfraz et al., 2023b) and ESMER (Sarfraz et al., 2023a). $\zeta$ refers to 'Activation Sparsity' in SCoMMER.

| Dataset | Buffer Size | SCoMMER | | | | | | ESMER | | | | | |
|---|---|---|---|---|---|---|---|---|---|---|---|---|---|
| | | $\zeta$ | $\eta$ | $\pi_h$ | $\pi_s$ | $\gamma$ | $r$ | $\eta$ | $\alpha_l$ | $\beta$ | $\gamma$ | $\alpha$ | $r$ |
| S-MNIST | 200 | – | – | – | – | – | – | 0.03 | 0.99 | 0.9 | 2.0 | 0.99 | 0.2 |
| S-CIFAR-10 | 200 | 0.8 | 0.1 | 0.5 | 2.0 | 0.15 | 0.5 | 0.03 | 0.99 | 1.2 | 0.15 | 0.999 | 0.1 |
| S-CIFAR-100 | 200 | 0.9 | 0.1 | 0.5 | 3.0 | 0.15 | 0.2 | 0.03 | 0.99 | 1.0 | 0.15 | 0.999 | 0.2 |
| S-Tiny-ImageNet | 200 | 0.9 | 0.05 | 0.5 | 3.0 | 0.1 | 0.1 | 0.03 | 0.99 | 2.5 | 0.15 | 0.999 | 0.07 |
| P-MNIST | 200 | – | – | – | – | – | – | 0.2 | 0.99 | 0.9 | 1.0 | 0.9 | 0.2 |
| R-MNIST | 200 | – | – | – | – | – | – | 0.2 | 0.99 | 0.9 | 1.0 | 0.99 | 0.2 |

model accumulates and consolidates more knowledge of recent and new tasks. (3) The working model learns more knowledge of new tasks. These results more clearly verify the effectiveness of our BDT-SMT, which can acquire more new knowledge from new tasks while consolidating the old knowledge. More concretely, we can observe the performance of the final obtained model (stable model) on learned tasks (i.e., the last row in the figure): firstly, the stable model can maintain a better classification performance on earlier tasks, indicating that the old knowledge can be better consolidated by the proposed BDT strategy; Secondly, the stable model achieves a good classification performance on recent and new tasks, indicating that the new knowledge can be better represented by the proposed SMT mechanism. These results show that our method can generalize well on both old and new tasks due to the combination of BDT and SMT. The combination of two components brings great benefits for continual learning, which enables the model to maintain a better balance between old and new knowledge to further improve model performance.

### A.5 HYPERPARAMETER SETTINGS

**Best Values** Table 10 shows the optimal hyperparameter settings of our BDT-SMT and SMT-Mean (i.e., single semantic memory model). In the table, $bs$ denotes the batch size, $\gamma$ denotes the balancing parameter in the loss function, $\eta$ denotes the learning rate, $f_S$ and $f_P$ denote the frequency parameter of stable model and plastic model, $f$ denotes the frequency parameter of single semantic memory model in SMT-Mean, and $\hat{m}$ denotes the forward transport momentum parameter. The selection of these hyperparameters is conducted with the backward transport momentum parameter $m_i (i \in P, S)$, minibatch size and threshold parameter $k$ fixed. The value of backward transport momentum parameter is the same as the corresponding forward transport momentum parameter for each dataset. The minibatch sizes are as follows: 128 for S-MNIST/P-MNIST/R-MNIST and 32 for S-CIFAR-10/S-CIFAR-100/S-Tiny-ImageNet. The threshold parameter $k$ is set to $\frac{1}{3}$ for all datasets.

Table 12: Hyperparameter ('p') values choice of our BDT-SMT and SMT-Mean.

| Method | p | S-MNIST | S-CIFAR-10 | S-CIFAR-100 | S-Tiny-ImageNet | P-MNIST | R-MNIST |
|---|---|---|---|---|---|---|---|
| SMT-Mean | $\gamma$ | [1.5, 2.0] | [0.1, 0.15, 0.2] | [0.1, 0.15, 0.2] | [0.1, 0.15, 0.2] | – | – |
| | $\eta$ | [0.03, 0.05, 0.1] | [0.05, 0.1, 0.15] | [0.03, 0.05, 0.1] | [0.03, 0.05, 0.1] | – | – |
| | $f$ | [0.6, 0.9, 1.0] | [0.2, 0.3, 0.6] | [0.1, 0.2, 0.5] | [0.04, 0.06, 0.08] | – | – |
| | $\hat{m}$ | [0.99, 0.999] | [0.99, 0.999] | [0.99, 0.999] | [0.99, 0.999] | – | – |
| BDT-SMT | $\gamma$ | [1.5, 2.0] | [0.1, 0.15, 0.2] | [0.1, 0.15, 0.2] | [0.1, 0.15, 0.2] | [1.0, 1.5] | [0.5, 0.75, 1.0] |
| | $\eta$ | [0.03, 0.1, 0.2] | [0.03, 0.1, 0.2] | [0.03, 0.1, 0.3] | [0.03, 0.05, 0.1] | [0.1, 0.2] | [0.1, 0.2] |
| | $f_S$ | [0.5, 0.9] | [0.1, 0.3, 0.5] | [0.1, 0.3, 0.5] | [0.02, 0.04, 0.06] | [0.5, 0.8] | [0.9, 1.0] |
| | $f_P$ | [0.8, 1.0] | [0.3, 0.6, 0.8] | [0.3, 0.6, 0.8] | [0.06, 0.08, 0.10] | [0.8, 1.0] | [1.0] |
| | $\hat{m}$ | [0.99, 0.999] | [0.9, 0.99, 0.999, 0.9999] | [0.99, 0.999] | [0.99, 0.999] | [0.99, 0.999] | [0.99, 0.999] |

Moreover, for the experiments of SCoMMER (Sarfraz et al., 2023b)/ESMER (Sarfraz et al., 2023a) on the datasets S-CIFAR-100 and S-Tiny-ImageNet, the best hyperparameter values are selected by fine-tuning the hyperparameters provided in the released code. This is due to the fact that these methods only provide hyperparameter values for shorter sequential tasks (i.e., $H = 5$ for S-CIFAR-100), while these parameters perform poorly when transferred to longer sequential tasks (e.g., $H = 10$). The detailed hyperparameter settings of these two methods are presented in Table 11. As for the hyperparameter settings of other comparative methods can be seen in our released code.

**Values Choice** In Table 12, we provide a list of hyperparameter values for selecting the best values in the grid-search strategy of our BDT-SMT/SMT-Mean (the buffer size is 200). For fair comparisons, we keep the experimental settings as close as possible to the CLS-ER (Arani et al., 2022), mainly adjusting the hyperparameters $\gamma, \eta, f_S, f_P, \hat{m}$. Note that in addition to the hyperparameter list we provide, there may be other better hyperparameter values for the model.

