# OpenReview forum: "Diving into Class-Incremental Learning from Better Balancing Old and New knowledge"
_ICLR.cc/2024/Conference — ICLR 2024 Conference Withdrawn Submission_

### Official Review · Reviewer_nG1u · 2023-10-22

**Soundness:** 3 good
**Presentation:** 2 fair
**Contribution:** 1 poor
**Rating:** 3
**Confidence:** 5

**Summary:**

This paper tackles the class-incremental learning problem from the Complementary Learning System (CLS) theory. Class-incremental learning is of great importance to the machine learning field. The proposed method is a combination of EMA and SI, which averages the model parameters among three models (working model, short-term model, and long-term model). The EMA process is combined with SI to weigh the importance of each parameter. The proposed method is evaluated on several datasets against other baseline methods.

**Strengths:**

1. Class-incremental learning is of great importance to the machine learning field.
2. The proposed method is evaluated on several datasets against other baseline methods.
3. This paper is easy to follow.

**Weaknesses:**

1. The novelty of the proposed method is not very strong. The current method somehow combines EMA and SI, making it incremental. The idea of using the working model and memory model is also similar to [1]. Although the idea of short-term memory and long-term memory is interesting, these models are utilized similarly in the updating process (e.g., Eq. 1 and Eq. 2). I may assume a complex learning system that includes the long-term recovery and short-term memory after reading the abstract, while the methodology is not so compatible with the “great CLS theory.”
2. Although using more parameters, the performance does not show substantial improvement. The current method requires at most three times a memory scale than the vanilla baseline. However, it shows little improvement to other baselines, given the extra budget. Besides, there is a similar work [2] that only uses at most two times the memory budget, which should be compared to in the experiments.
3. Too many hyper-parameters make the model fragile. Eq. 1 includes two hyper-parameters for the thresholding. As shown in Table 10, the choice of these parameters differs substantially from dataset to dataset, making the model fragile. Moreover, three more hyperparameters in the table need to be grid searched.
4. Lacking comparison on large-scale datasets. It is essential to perform the comparison on ImageNet1000 and show the ability of class-incremental learning algorithms with large-scale inputs.
5. Unclear points. The total loss in Eq. 6 is a combination of cross-entropy and consistency. What does sf mean in the cross-entropy loss? Besides, the description of consistency loss is also ambiguous. What is the “optimal strategy”? If the consistency loss is the MSE loss, why not show it with notations? These unclear points make the paper half-baked and less convincing.

[1] Error Sensitivity Modulation based Experience Replay: Mitigating Abrupt Representation Drift in Continual Learning. ICLR 2023

[2] Foster: Feature boosting and compression for class-incremental learning. ECCV 2022

**Questions:**

Please refer to the weaknesses part and address them during rebuttal.

In summary, although this paper addresses an important problem, the incremental contribution, half-baked evaluations, and complex hyper-parameters are my main concerns.

---

### Official Review · Reviewer_W1qo · 2023-10-27

**Soundness:** 3 good
**Presentation:** 2 fair
**Contribution:** 1 poor
**Rating:** 3
**Confidence:** 4

**Summary:**

This paper focuses on the representation learning of old and new knowledge in Class-IL. It proposes a bidirectional
transport strategy to transfer parameters directly and bidirectionally between the working model and the two semantic memory models and a selective momentum mechanism to selectively update parameters based on the parameter importance.

**Strengths:**

1. The figures for illustrating the idea are clear.
2. This paper varifies their method's effectiveness on four datasets and seems that the results are promising.

**Weaknesses:**

(1) The writing is poor. In the introduction, you should explain your motivation first and then introduce your method. The details of various continual learning methods should be put in the related works.

In section 2.3, the emprical result of the BDT Strategy should be put in the experiment section. The structure of this paper is very unclear.

(2) The method contribution is incremental. The parameter selection strategy is not novel. Also, why transferring previous knowledge to the learning of the current task can reduce the average forgetting?

(3) Putting the related section into Appendix is not a good idea. You can cut some unimportant content in this paper.

**Questions:**

Please revise your paper again and again. The structure of this paper is very unclear and you can cut many places to reduce the paper length.

---

### Official Review · Reviewer_UGyp · 2023-10-28

**Soundness:** 3 good
**Presentation:** 2 fair
**Contribution:** 2 fair
**Rating:** 6
**Confidence:** 4

**Summary:**

This paper proposes a new training approach for class-incremental learning, named BDT-SMT, based on the idea of Complementary Learning System (CLS). In the training pipeline, three models, i.e., a working model, a stable model, and a plastic model, are held to balance the trad-off between preserving previous learned old knowledge and learning new knowledge. Through the proposed Bidirectional Transport (BDT) Strategy, the three models can transport their own current knowledge to each other, ensuring the knowledge sharing between old tasks and new tasks. Moreover, the proposed Selective Momentum (SMT) Mechanism helps determine which parameters of the working model should be updated. Experimental results on four datasets show that the proposed method achieves best performance among all baselines.

**Strengths:**

1. The proposed Bidirectional Transport (BDT) is a novel and interesting training strategy for class-incremental learning based on momentum update, with which the three models can share their own knowledges to not only preserving working model from forgetting of old knowledge, but also enable the semantic memory models to update their learned old knowledge.

2. The proposed Selective Momentum (SMT) seems to be helpful for selecting parameters to update for combining old knowledge and new knowledge.

3. Experimental results are promising and solid.

**Weaknesses:**

While this paper proposes a interesting approach for balancing old and new knowledge in class-incremental learning, I have some concerns as follows:
+ In the Semantic Memory Models, the only difference between the stable model and the plastic model is their momentum update parameters and update frequency. This could be trivial for distincting the these two models. The key idea here is to make the stable model preserve more general information among all tasks and make the plastic model retain more recent task's information, but with this trivial difference, it is hard to say the goal has been achieved. Thus I am not sure what are the actual roles of them in this architecture. Further, the author also need to provide more insight for holding two Semantic Memory Models, that's say, why only one semantic memory model is not enough to hold old knowledge?
+ Some training details should be further clarified. For example, for the process of (1) momentum update of Semantic Memory Models (Eq. 1), (2) momentum update of working model (Eq. 2), I wonder, after back-propogating update the working model, should first do (1) and then (2)? or reversed？
+ During testing after the training of each incremental task. Which model is used for inference?

**Questions:**

See above

---

### Official Review · Reviewer_T7Kt · 2023-10-30

**Soundness:** 2 fair
**Presentation:** 3 good
**Contribution:** 2 fair
**Rating:** 3
**Confidence:** 4

**Summary:**

This paper proposed BDT-SMT (bidirectional transport - selective momentum) based on the Complementary Learning System theory. Specifically, the BDT module integrates the knowledge of new and old tasks in the working model, while the SMT module chooses the parameters that are unimportant to the old tasks for the working model updating.

**Strengths:**

+ This paper proposes a new way of information transfer from the semantic models to the working model, which can improve the performance of the previous CLS-based method.
+ Extensive experiments show the effectiveness of the proposed method. Moreover, the ablation study demonstrates the strengths of both BDT and SMT.
+ Most parts of this paper are well written and easy to understand.

**Weaknesses:**

+ The motivation of BDT is not clear. The author claims that the previous methods only concerned one unidirectional process (working model  semantic memory models). However, during updating the working model, the previous method also contains the knowledge distillation from the semantic memory models to the working model (i.e., an MSE loss between the logits of the working model and semantic memory models), which is also a type of so-called forward transport. The proposed BDT in this paper is an additional way of forward transport.
+ Besides, the author mentioned that ‘the representation of new knowledge is often impaired due to the excessive integration of the retained old knowledge under the Class-IL setting’. However, there is no explanations or experiments to illustrate how the excessive integration of old knowledge degrades new knowledge, and how the proposed method alleviates the excessive integration.
+ The author does not show the forgetting measure of the compared methods. From Fig. 3, comparing ERMER, the proposed method seems to tend to sacrifice accuracy on the old tasks to achieve a better performance on the new task. This obeys the spirit of continual learning, which mainly hopes to alleviate forgetting rather than only improve average accuracy.
+ To calculate the parameter importance, SMT has to store parameters of the old working model after the training on the last task, which enlarges the requirements of the model storage and is unfair to other comparison methods.
+ The author does not explain how to choose the stable model, plastic model, both of them or even none of them in Eq. (2).
+ From Alg. 1, it seems like a two-stage approach, where the first stage is to update the working model by SGD, and the second stage is to fine-tune the working model by SMT. It would be better if the author could provide a specific and detailed algorithm or pseudo code about BDT-SMT.

**Questions:**

Please refer to the weakness section.

Additional question: There are many concerns about the parameter importance methods like SI used in SMT. The important/unimportant parameters of the working model, stable model, and plastic model may not be the same. So the unimportant parameters of the working model may be changed to unimportant parameters of the stable or plastic model, which is unclear and hard to analyze.